# Nutritional Status Is Associated with Function, Physical Performance and Falls in Older Adults Admitted to Geriatric Rehabilitation: A Retrospective Cohort Study

**DOI:** 10.3390/nu12092855

**Published:** 2020-09-18

**Authors:** Miriam Urquiza, Naiara Fernandez, Ismene Arrinda, Irati Sierra, Jon Irazusta, Ana Rodriguez Larrad

**Affiliations:** 1Department of Physiology, University of the Basque Country, UPV/EHU, 48940 Leioa, Spain; jon.irazusta@ehu.eus (J.I.); ana.rodriguez@ehu.eus (A.R.L.); 2Geriatric Department, Igurco Servicios Socio Sanitarios, Grupo IMQ, 48011 Bilbao, Spain; nfernandez@igurco.es (N.F.); iarrinda@igurco.es (I.A.); isierra@igurco.es (I.S.)

**Keywords:** nutritional status, geriatric rehabilitation, functional status, physical performance, fallers

## Abstract

Nutritional status is relevant to functional recovery in patients after an acute process requiring rehabilitation. Nevertheless, little is known about the impact of malnutrition on geriatric rehabilitation. This study aimed to determine the association between nutritional status at admission and the evolution of functional and physical outcomes, as well as the capability of nutritional status to identify fallers among patients admitted to geriatric rehabilitation for different reasons. This was a retrospective cohort study of 375 patients. Data collected included age, gender, diagnosis at admission, comorbidities, cognitive and nutritional status, functional and physical measurements, length of stay, mortality and falls. Orthogeriatric patients with worse nutritional status according to the Mini Nutritional Assessment-Short Form (MNA-SF) had a significantly lower Barthel Index at admission and discharge with worse functional gain and poorer outcomes in the Short Physical Performance Battery (SPPB). However, in hospital-deconditioned patients, the MNA-SF score was not significantly associated with functional and physical recovery. Poor nutritional status at admission increased the risk of experiencing at least one fall during rehabilitation in orthogeriatric patients. However, hospital-deconditioned patients who fell had better SPPB scores than those who did not fall. Our results demonstrate the importance of nutritional status in the clinical evolution of orthogeriatric patients throughout the rehabilitation process.

## 1. Introduction

In older adults, hospitalization due to an acute illness is often associated with increased functional and cognitive decline [1,2]. Approximately 35% of older patients are discharged with worse performance in activities of daily living than before being hospitalized [3]. Consequently, after hospitalization, some older adults are temporarily admitted to an in-patient geriatric rehabilitation ward with the aim of recovering their functional and physical status so that they can return to their homes.

Geriatric rehabilitation, defined by the European Consensus on Geriatric Rehabilitation, provides temporal integral care for older patients following hospitalization [4]. Although there are large differences between countries regarding the structure and delivery of geriatric rehabilitation, care is usually administered by a multidisciplinary team consisting of at least a physician trained in geriatric rehabilitation, a physiotherapist and a nurse [4]. Patients admitted to these units include orthogeriatric patients (i.e., patients who have experienced hip fracture, knee replacement or polytraumatism), hospital-deconditioned patients (i.e., patients who have been hospitalized for exacerbation of chronic heart failure, neurologic patients (i.e., patients who have undergone a stroke) and others (i.e., amputees). Rehabilitation usually starts with a comprehensive geriatric assessment with a multidisciplinary approach for the care of older patients [5,6]. Furthermore, in-patient rehabilitation services designed for older adults have beneficial effects on functional and cognitive improvement, prevention of institutionalization and reduction of mortality [7,8,9].

Increasing evidence suggests that in older adults, nutritional status is relevant for recovery from an acute illness [10]. Moreover, poor nutritional status during hospitalization is associated with longer hospitalization and increased mortality rates [11] and is related to adverse outcomes in a wide variety of acute conditions such as hip fractures [12] and stroke [13]. Consequently, early identification of malnutrition in geriatric rehabilitation is especially important since it facilitates nutritional interventions that may help to optimize clinical outcomes. There are different nutritional screening tools directed at older populations, such as nutritional risk screening, mini nutritional assessment (MNA), malnutrition screening test and subjective global assessment [14,15]. All of these tools are validated and widely used. However, the European Society for Clinical Nutrition and Metabolism (ESPEN) recommends the MNA, either in its full or short forms, since it also captures physical and mental aspects that frequently affect the nutritional status of the elderly [16]. The short form of the MNA (MNA-SF) is an easy and rapid nutrition screening tool with good accuracy for assessing malnutrition and risk of malnutrition in older people in different healthcare settings [17].

The prevalence of malnutrition in older adults is different between healthcare settings and is positively correlated with the level of dependence associated with each care setting as well as the degree of cognitive impairment and number of chronic diseases experienced by the patients [18]. The lowest malnutrition levels are reported in community-dwelling older adults (<5%), and the highest malnutrition levels are reported in rehabilitation and sub-acute care patients (between 20–36%, depending on the assessment tool used) [19,20,21].

A recent review analyzing nutritional status in geriatric rehabilitation patients reported that malnutrition was negatively associated with functionality, emphasizing the necessity of screening for malnutrition in this population [22]. Indeed, other studies in these settings have found higher risk of adverse outcomes, such as increased length of stay and mortality, in patients with malnutrition [23,24,25]. Cross-sectional studies in ambulatory geriatric [26] and home care settings [27] also associated deficient nutritional status with low physical performance. However, the predictive value of nutritional status on the clinical evolution of geriatric patients throughout the rehabilitation process is poorly understood.

Malnutrition is also related to higher incidence of falls and risk of falling in hospitalized and community-dwelling older adults [28,29]. These events can lead to many adverse consequences in this population, such as functional decline, increased morbidity and even death [30,31]. However, little is known about relationship between malnutrition and risk of falling in geriatric rehabilitation settings, although low functional and physical performance after hospitalization-deconditioning [32] may directly increase this risk.

Furthermore, ESPEN classifies malnutrition depending on its etiology. In patients with acute conditions, such as hip fractures, malnutrition is usually injury-related and is characterized by acute pro-inflammatory activity and a fast decline of body energy and nutrient stores. In contrast, chronic disease-related malnutrition, such as chronic heart failure, combines on-going systemic inflammation with chronic weight loss and cachexia [16]. Consequently, the characteristics of malnutrition may be different depending on the reason for admission to the geriatric rehabilitation ward.

With this background, the first aim of this study was to determine the association between nutritional status at admission and the evolution of functional outcomes and physical performance during an in-patient geriatric rehabilitation process. Secondly, we aimed to analyze the capability of nutritional status at admission to identify patients who are at higher risk of experiencing at least one fall during the period of ward stay. Furthermore, taking into account the different profiles of patients admitted to rehabilitation units, we aimed to clarify whether these associations depend on the reason for the patient’s admission. We hypothesize that worse nutritional status at rehabilitation admission is associated with worse evolution of functional and physical outcomes and with higher risk of being a faller.

## 2. Materials and Methods

### 2.1. Study Design, Setting and Participants

This was a retrospective cohort study of the medical records of patients admitted to the Igualatorio Médico Quirúrgico (IMQ) Igurco Orue geriatric in-patient rehabilitation ward. From September 2015 to December 2019, 597 patients were admitted to the ward. Inclusion criteria of our analysis were older adults aged ≥ 65 years who fully participated in the rehabilitation program and were assessed at least at admission and before discharge. Palliative care patients were excluded. After screening medical histories and inclusion criteria, 375 patients were eligible for analysis (Figure 1). According to their diagnosis at admission, 239 (63.8%) patients were orthogeriatric, 104 (27.7%) were admitted due to hospital-associated deconditioning and 32 (8.5%) were admitted due to other conditions (i.e., neurologic events or amputation). The study was approved by the Committee on Ethics in Research of the University of the Basque Country (Code number: M10/2019/198).

The rehabilitation ward was staffed by full-time geriatricians, 24 h nurses, nursing assistants, physical therapists and psychologists, and focused on functional recovery of older adults after an acute illness and hospitalization, with the aim to return them to their previous dwelling situation. The rehabilitation program consisted of individualized one-hour rehabilitation sessions 5 days/week throughout the period of admission (Table 1). 

Nutritional intervention in patients with malnutrition or at risk of malnutrition began at admission according to Spanish Society of Geriatrics and Gerontology (SEGG) guidelines [33]. The nutritional intervention protocol consisted of diet optimization (i.e., increasing protein intake) and/or nutritional supplementation. Supplements were prescribed according to the requirements and co-morbidities of the patient.

### 2.2. Data Collection and Outcomes

Participant data were retrieved from the geriatric rehabilitation ward database, and included demographic characteristics (age, sex) and diagnosis at admission. According to clinical data, comorbidity was assessed by the age-adjusted Charlson comorbidity index (ACCI), adding one point for each decade over 50 years of age [34]. 

Nutritional status data were collected at patient admission by the MNA-SF [17]. The MNA-SF test is a validated screening tool to identify malnourished older adults or those at risk of malnutrition. The test has two versions depending on whether calf circumference or body mass index is used for the anthropometric measurement. In this study, we used the body mass index item. The test provides a maximum score of 14 points. A total score of 12–14 points indicates that the patient is well-nourished, 8–11 points indicates risk of malnutrition, and 0–7 points indicates malnutrition. 

Cognitive status was assessed by the Spanish version of the Mini Mental State Examination (MMSE). The MMSE test is a cognitive function test based on a total possible score of 30 points that includes orientation tests, concentration, attention, verbal memory, naming and visuospatial skills [35]. A higher score indicates better cognitive status.

Functional status was assessed by the Barthel index (BI), which measures older patients’ dependence level in activities of daily living [36]. The BI comprises 10 items with total scores ranging from 0 points (worst independence level) to 100 points (full independence in activities of daily living). The BI score was collected at patient admission and discharge. The functional outcomes of the rehabilitation process related to BI score were the absolute functional gain (AFG), defined as the difference between the BI score at discharge and the BI score at admission (AFG = BI at discharge−BI at admission) and the relative functional gain (RFG) that shows the percentage of functional capacity recovered at discharge. The RFG was defined as the AFG divided by the difference between maximum BI score (100 points) and BI score at admission (RFG = AFG/(100−BI at admission)) [37]. Higher scores in the AFG and RFG indicate better functional gain.

Physical function was assessed by the following three performance tests at admission and discharge: Short Physical Performance Battery (SPPB), the Tinetti Performance-Oriented Mobility Assessment (POMA) scale and Holden’s Functional Ambulation Category (FAC). 

SPPB is a battery of three tests that combines assessment of balance, gait speed and lower limb strength. Each test is scored on a scale of 0–4 points, with a total performance score range of 0–12 points using cut-off point criteria established by Guralnik et al. [38]. Higher scores indicate better physical function. The Tinetti POMA test is a reliable and valid clinical test used to measure balance and gait in older people. The total Tinetti-POMA scale comprises balance (POMA-B) and gait subscales (POMA-G). The maximum score is 16 points for POMA-B and 12 points for POMA-G, with a maximum possible total Tinetti-POMA score of 28 points [39]. The FAC is an easy method for classifying mobility. The FAC has 6 categories ranging from 0 (non-functional ambulation) to 5 (independent). The intermediary points quantify levels of assistance for walking, supervision or independent but limited mobility [40].

Length of stay in days (LOS), discharge setting, number of falls and mortality were also collected. Every fall was reported and recorded in the electronic database by nurses. Finally, researchers extracted the data from the electronic system. 

### 2.3. Statistical Analysis 

Continuous variables were expressed as means with standard deviations (SD), and categorical variables were expressed as frequency counts and percentages (%). The Kolmogorov–Smirnov test was used to assess the normality of distribution. One-way ANCOVA was conducted to determine the differences in physical and functional status between patients with different nutritional statuses at admission. The analysis was controlled for ACCI, physical and functional status at admission and for ACCI, admission values and length of stay at discharge. Partial eta-squared (η*_p_*^2^) was calculated to estimate the effect size. η*_p_*^2^ values of 0.01, 0.06, and >0.138 were considered small, medium and large, respectively [41]. Participant characteristics at admission were also compared between those who experienced at least one fall during the rehabilitation process (fallers) and those who did not (non-fallers) by Student´s T test (normally distributed data) or the Mann–Whitney U-test (non-normally distributed data). Variables with *p* < 0.1 in univariate analysis were considered eligible for a logistic multivariate regression model to identify variables independently associated with being a faller. The Hosmer–Lemershow test was used to determine the goodness-of-fit of the model and thus to determine if the observed event rate matched the expected one. A number closer to 1 showed a better goodness-of-fit. Omnibus was used to test whether the explained variance was significantly greater than the unexplained variance. Nagelkerke’s *R*^2^ value estimated the proportion of the dependent variable explained by the independent variables. Missing values were not included in the analysis. Statistical analysis was performed in the entire sample and according to diagnosis at admission in orthogeriatric and hospital-deconditioned groups. Amputees and neurological patients were not analyzed independently due to the low number of participants in these groups. In all tests, differences were considered significant at *p* < 0.05. Statistical analysis was performed using the IBM Corp. Released 2012. IBM SPSS Statistics for Windows, Version 21.0. Armonk, NY: IBM Corp.

## 3. Results

### 3.1. Study Participants

Orthogeriatric patients were significantly older than hospital-deconditioned patients and were more likely to be females (*p* < 0.001) (Table 2). However, hospital-deconditioned patients had significantly more comorbidities (*p* = 0.034) and poorer nutritional status at admission (*p* = 0.029) than orthogeriatric patients. There were no well-nourished patients at admission, whereas 164 (43.7%) were malnourished and 211 (56.3%) were at risk of malnutrition. In the orthogeriatric group, 97 (40.6%) patients were malnourished, whereas in the hospital-deconditioned group, 54 (51.9%) patients were malnourished. There were no differences between groups in BI at admission, cognitive status, falls, LOS or discharge setting (*p* > 0.05).

### 3.2. Functional Outcomes According to Nutritional Status

In the whole sample, ANCOVA revealed significant differences between patients with malnutrition and those at risk of malnutrition in all functional variables. Thus, BI at admission (F(1372) = 22.23, *p* < 0.001), BI at discharge (F(1322) = 6.18, *p* = 0.013), AFG (F(1322) = 6.18, *p* = 0.013) and RFG (F(1322) = 5.71, *p* = 0.017) were lower in malnourished patients than in those at risk of malnutrition (Table 3). 

When stratified for diagnosis at admission, differences in functional outcomes between orthogeriatric patients with malnutrition and at risk of malnutrition remained significant. Therefore, BI at admission (F(1236) = 8.32, *p* = 0.004), BI at discharge (F(1209) = 5.21, *p* = 0.023), AFG (F(1209) = 5.21, *p* = 0.023) and RFG (F(1209) = 8.13, *p* = 0.005) were significantly lower in malnourished patients compared with those at risk of malnutrition. Similarly, in hospital-deconditioned patients, BI at admission was significantly lower in malnourished patients compared to patients at risk of malnutrition (F(1101) = 4.75, *p* = 0.032). However, in this group, there were no differences between groups in BI at discharge, AFG and RFG (*p* > 0.05 for all variables) (Table 3).

### 3.3. Physical Performance According to Nutritional Status

In the whole sample, malnourished patients had significantly worse physical performance at admission in SPPB (F(1354) = 11.05, *p* = 0.001), POMA (F(1372) = 7.07, *p* = 0.008) and FAC tests (F(1372) = 6.53, *p* = 0.011) than those at risk of malnutrition. Malnourished patients also had significantly worse physical outcomes at discharge in SPPB (F(1303) = 8.49, *p* = 0.004), and POMA tests (F(1323) = 4.05, *p* = 0.045) than those at risk of malnutrition. However, there were no differences between groups in the FAC test at discharge (*p* > 0.05) (Table 4).

Malnourished orthogeriatric patients had significantly worse physical performance at admission in SPPB (F(1218) = 7.92, *p* = 0.005), POMA (F(1236) = 4.97, *p* = 0.027), and FAC tests (F(1236) = 5.26, *p* = 0.023) than those at risk of malnutrition. At discharge, malnourished orthogeriatric patients also had significantly lower SPPB (F(1101) = 8.63, *p* = 0.004) scores than orthogeriatric patients at risk of malnutrition, whereas no significant differences were found in POMA and FAC tests between groups (*p* > 0.05 for both variables). Finally, in hospital-deconditioned patients, no significant differences were found between nutritional status groups in physical performance at admission and discharge (*p* > 0.05 for all variables) (Table 4).

### 3.4. Identification of Fallers during the Rehabilitation Process

A univariate analysis was conducted to determine differences at admission in clinical, cognitive, functional and physical characteristics between patients who fell and did not fall throughout the rehabilitation process. In the whole sample, patients who fell had significantly poorer nutritional and cognitive statuses (*p* = 0.003; *p* = 0.005, respectively) than those who did not fall. Among orthogeriatric patients, fallers had higher ACCI scores (*p* = 0.044), lower MNA-SF scores (*p* < 0.001), lower MMSE scores (*p* = 0.001) and worse SPPB scores at admission (*p* = 0.019) than non-fallers. Finally, hospital-deconditioned patients who suffered at least one fall had significantly higher scores on the SPPB (*p* = 0.022), POMA (*p* = 0.007) and FAC tests (*p* = 0.025) at admission than non-fallers (Table 5). When variables with *p* < 0.1 in the univariate analysis between patients who fell and did not fall were introduced in a multivariate regression logistic model, lower nutritional status at admission was a significant factor for identifying fallers in the whole sample (OR: 0.807; 95% CI: 0.696–0.936) and in orthogeriatric patients (OR: 0.806; 95% CI: 0.657–0.990). In the whole sample, higher ACCI also remained significantly associated with fallers (OR: 1.148; 95% CI: 1.010–1.304), whereas in orthogeriatric patients, poor cognitive status was significantly associated with increased risk of being a faller (OR: 0.938; 95% CI: 0.892–0.987). However, in hospital-deconditioned patients, higher SPPB scores at admission were observed in those who fell throughout the rehabilitation process (OR: 1.236; 95% CI: 1.052–1.452) (Table 6).

## 4. Discussion

Our findings showed that, in orthogeriatric patients, malnutrition at admission was associated with worse evolution of functional and physical outcomes throughout the rehabilitation process in a geriatric ward. Likewise, poor nutritional status, together with a low MMSE score, was an independent risk factor for being a faller. Nevertheless, in hospital-deconditioned patients, the MNA-SF score at admission did not correlate to the evolution of functional and physical outcomes throughout the rehabilitation process. In this group, a higher SPPB score is an independent risk factor for being a faller.

The prevalence of malnutrition in the present study is higher than that reported in most healthcare settings [19,20,21]. In contrast, other studies that analyzed hospital-deconditioned older patients reported higher levels of malnutrition, similar to the present research [42,43]. This difference may be due to the tool used to define malnutrition. Studies with a malnutrition prevalence similar to that in our study used the MNA-SF. However, a lower percentage of malnutrition was found in studies that defined malnutrition using other tools. Notably, the MNA-SF has higher sensitivity but lower specificity than other scales [44]. This may explain the high prevalence of malnutrition in studies using the MNA-SF since the number of false positives is high, but the number of false negatives is low.

Our results also showed that, in the whole sample, poorer nutritional status at admission was associated with worse evolution of functional and physical outcomes and with increased risk of experiencing a fall throughout the rehabilitation process. However, this association depends on the reason for admission. In orthogeriatric patients, a low MNA-SF score was associated with worse evolution and increased risk of being a faller. In contrast, this association was not observed in hospital-deconditioned patients. This fact may be explained by the different characteristics and malnutrition etiology of hospital-deconditioned and orthogeriatric patients. Hospital-deconditioned patients are usually hospitalized after exacerbation of chronic diseases, and they often have many comorbidities and poor nutritional status, especially patients in advanced disease stages and those who are acutely decompensated [45,46]. A high number of comorbidities and poor nutritional status were also identified in the hospital-deconditioned patients analyzed in our study. In these patients, malnutrition could be long-term and related to the severity of the chronic disease with associated cachexia [47]. In fact, chronic disease patients are known to have chronically decreased muscle mass [48]. In contrast, orthogeriatric patients, who are usually admitted for a recent acute condition such as a hip fracture, could be more affected by acute changes in nutritional status [16]. Therefore, we hypothesize that chronic muscle mass loss and severity of symptoms related to chronic disease in hospital-deconditioned patients could modulate the impact of their MNA-SF score on functional and physical recovery. Other scales that measure illness severity, such as the full MNA and/or the assessment of cachexia or chronic inflammatory biomarkers could more accurately predict the clinical evolution of hospital-deconditioned patients in geriatric rehabilitation.

Our findings in orthogeriatric patients are in line with other studies, where nutritional status was associated with functional recovery [49,50]. In these studies, follow-up at 3 or 6 months by telephone interview also showed that in patients with poorer nutritional status, functional outcomes remained worse than in those with better status [49,50]. To our knowledge, this is the first study to analyze the association between malnutrition at admission and the evolution of physical performance throughout the rehabilitation process in orthogeriatric patients. In a cross-sectional study, Chevalier et al. showed that poorer nutritional status measured by the full MNA was associated with lower gait speed in patients undergoing ambulatory rehabilitation, but reasons for patient admission were not specified [26]. We observed that orthogeriatric patients with malnutrition at admission had significantly worse SPPB scores throughout the rehabilitation process than those at risk of malnutrition. Poorer physical outcomes in these patients could influence their reduced food intake or assimilation [16]. These factors could reduce protein availability for muscle and increase muscle catabolism [51], which could affect lean muscle gain and the positive evolution of physical outcomes throughout the rehabilitation process.

A lower MNA-SF score at admission was also associated with increased risk of being a faller in orthogeriatric patients. Our results also confirm the relationship between nutritional status and falls in orthogeriatric patients in other clinical settings [28,29]. A fall represents an adverse outcome that could influence the rehabilitation process with worse functional recovery, increasing the length of stay and reducing the probability of being discharged at home [52]. Although the tests that usually determine the risk of falling in this population, such as POMA–Tinetti or SPPB, were not included in the multivariable model for predicting patients who fell, the MNA-SF and MMSE scores remained significant in the last equation of the backward regression model. However, the capacity of the MMSE to identify patients at risk of falling agrees with previous studies demonstrating that cognitive impairment increases the risk of falls [53].

In contrast, we did not find differences in the evolution of any functional or physical outcomes during rehabilitation between malnourished hospital-deconditioned patients and those at risk of malnutrition. These results are in line with other studies, where the direct association between nutritional status and functional outcomes in hospital-deconditioned patients was unclear. Goto et al. [54] reported that a better MNA-SF score at the start of rehabilitation was only significantly associated with functional recovery in deconditioned patients who were previously more dependent for activities of daily living, while in more independent patients, better nutritional status was not an independent factor for predicting better functional outcomes. In contrast, Katano et al. [42] reported higher functional gain in patients with worse nutritional status (MNA-SF ≤ 7) than in those with better nutritional status (MNA-SF > 7). In hospital-deconditioned patients, poor nutritional status at admission was not associated with poorer physical recovery. As far as we know, there are no studies that analyze the evolution of physical performance of patients with different nutritional statuses in clinical settings. In a cross-sectional study carried out in deconditioned patients undergoing rehabilitation in a geriatric day hospital, lower nutritional status was not associated with poorer physical performance [55]. Furthermore, in another cross-sectional study carried out with moderate-to-severe chronic obstructive pulmonary disease (COPD) patients, poorer nutrition did not show an independent association with lower physical performance [56]. These results suggest that recovery of physical performance in hospital-deconditioned patients could be dependent on other factors not detected by the MNA-SF.

Surprisingly, in deconditioned patients, a higher score on the SPPB test at admission was associated with higher risk of being a faller. Contrary to our findings, lower SPPB has been associated with a higher risk of falls among in-hospital patients and community-dwelling older adults [57,58]. However, the SPPB score among hospital-deconditioned fallers in our study was remarkably lower than that in previous studies. We hypothesize that in our study, hospital-deconditioned patients with a higher score on the SPPB might be more independent for displacements and consequently, have a higher chance of experiencing falls than patients with poor physical performance. 

An implication of the present study for clinical practice is the usefulness of the MNA-SF nutrition screening tool for prediction of functional and physical outcomes in geriatric rehabilitation patients with acute orthogeriatric conditions. The MNA-SF is recommended by ESPEN for nutrition screening in older adults and is widely used in hospital and rehabilitation settings [14]. In fact, recently published ESPEN guidelines for clinical nutrition in older persons with malnutrition or at risk of malnutrition encourage nutritional supplementation as well as exercise in older adults with specific diseases following orthopedic surgery, especially for hip fractures [59], but no specific guidelines are described for hospital-deconditioned patients following rehabilitation.

A strength of this study is that it is the first to assess the relationship between nutritional status and both functional and physical performance outcomes in geriatric rehabilitation patients with different reasons for admission. Additionally, our findings indicate that malnutrition in orthogeriatric patients is a risk-factor for being a faller throughout the rehabilitation process. However, some limitations need to be addressed. First, the MNA-SF is a validated tool for screening of malnutrition, but use of the full MNA might have improved our study´s specificity, particularly in hospital-deconditioned patients. For instance, assessing anthropometric measurements such as mid-arm and calf circumferences and illness severity would further complete the nutritional status assessment. Second, we only assessed the MNA-SF at admission and did not reassess patient nutritional status throughout the process. Therefore, patients who modified their nutritional status during rehabilitation might have different outcomes. Further research is needed to clarify the effects of nutritional status and comprehensive nutritional and exercise interventions in this specific population.

## 5. Conclusions

Malnutrition was highly prevalent among older patients in the post-hospitalization rehabilitation ward. Poor nutritional status, measured by the MNA-SF, was associated with worse evolution of functional and physical outcomes throughout the rehabilitation process in orthogeriatric patients. In contrast, in hospital-deconditioned patients, a low MNA-SF score was not associated with worse evolution.

Malnutrition at admission in geriatric rehabilitation wards could help to identify orthogeriatric patients at higher risk of being a faller. In hospital-deconditioned patients, a higher score on the SPPB was an independent factor for identifying fallers. However, because SPPB scores are especially low at admission in these patients, our results cannot be extrapolated to other types of patients. Our results demonstrate the importance of nutritional status in the clinical evolution of orthogeriatric patients.

## Figures and Tables

**Figure 1 nutrients-12-02855-f001:**
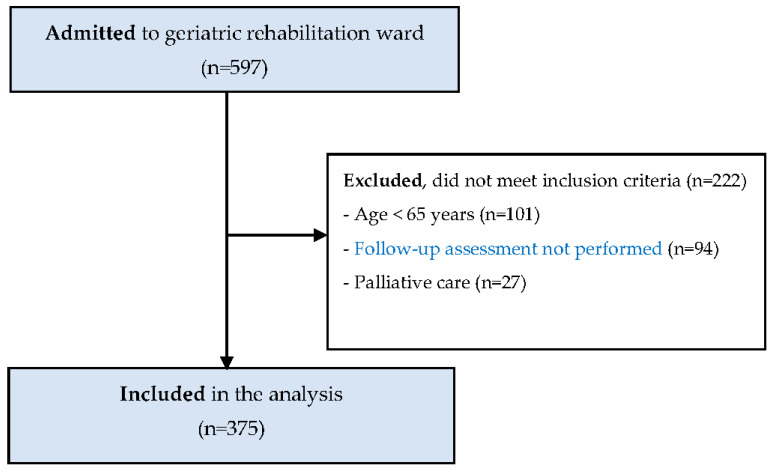
Study flow diagram.

**Table 1 nutrients-12-02855-t001:** Rehabilitation program.

Rehabilitation Program	
**Joint mobilizations/manual therapy**	
**Upper limb strength**	Exercises with a pulley
**Lower limb strength**	Knee extension exercises (personalized load)
	Standing exercises (hip flexion, abduction and extension)
	Chair stand exercises
**Balance**	Side-by-side stand
	Semi-tandem stand
	Tandem stand
**Gait reeducation**	Adapted parallel bars walking
	Walking with obstacles
	Gait retraining with different assistance devices (i.e., walkers or canes)

**Table 2 nutrients-12-02855-t002:** Participant characteristics.

	Total Sample *n* = 375	Orthogeriatric *n* = 239	Hospital-Deconditioned *n* = 104	*p*
Age (years)	82.9 ± 6.97	84.07 ± 6.79	80.77 ± 6.67	<0.001
Female sex, *n* (%)	256 (68.3)	184 (77)	60 (57.7)	<0.001
ACCI	5.93 ± 1.87	5.74 ± 1.87	6.14 ± 1.83	0.034
Barthel Index	32.43 ± 23.03	32.30 ± 21.46	34.42 ± 25.15	0.072
MNA-SF	7.43 ± 1.62	7.54 ± 1.65	7.16 ± 1.52	0.029
Malnutrition, *n* (%)	164 (43.7)	97 (40.6)	54 (51.9)	0.052
Cognitive status, *n* (%)				0.741
No CI	152 (40.6)	99 (41.4)	44 (42.3)	
Mild CI	104 (27.8)	64 (26.8)	31 (29.8)	
Moderate CI	69 (18.4)	46 (19.2)	15 (14.4)	
Severe CI	49 (13.1)	30 (12.6)	14 (13.5)	
Fallers, *n* (%)	101 (26.9)	62 (25.9)	29 (27.9)	0.708
LOS (days)	89.88 ± 48.58	88.95 ± 47.3	91.04 ± 50.63	0.870
Mortality, *n* (%)	48 (12.8)	25 (10.5)	18 (17.3)	0.078
Discharge, *n* (%)				0.097
Home discharge	240 (73.8)	165 (77.5)	58 (68.2)	
Institutionalized	85 (26.2)	48 (22.5)	27 (31.8)	

Abbreviations: ACCI: Age-adjusted Charlson comorbidity index; MNA-SF: Mini Nutritional Assessment Short Form; CI: Cognitive impairment; LOS: Length of stay.

**Table 3 nutrients-12-02855-t003:** Functional outcomes according to nutritional status.

	Malnourished	Risk of Malnutrition	ANCOVA ^1^	Effect Size (η_p_^2^)
**Total sample**				
BI at admission	25.80 ± 21.74	37.58 ± 22.72	<0.001	0.056
BI at discharge	63.46 ± 27.18	76.06 ± 23.43	0.013	0.019
Absolute functional gain	35.28 ± 24.91	37.02 ± 23.34	0.013	0.019
Relative functional gain	49.35 ± 48.19	62.06 ± 33.88	0.017	0.017
**Orthogeriatric**				
BI at admission	26.57 ± 20.11	36.21 ± 21.53	0.004	0.034
BI at discharge	63.80 ± 25.48	76.95 ± 24.07	0.023	0.024
Absolute functional gain	34.98 ± 27.35	39.22 ± 23.05	0.023	0.024
Relative functional gain	45.54 ± 55.93	65.26 ± 33.39	0.005	0.037
**Hospital-Deconditioned**				
BI at admission	29.35 ± 24.06	39.90 ± 25.39	0.032	0.045
BI at discharge	68.60 ± 26.24	75.05 ± 23.12	0.912	<0.001
Absolute functional gain	37.19 ± 18.08	33.28 ± 23.18	0.912	<0.001
Relative functional gain	60.43 ± 29.56	57.95 ± 33.13	0.318	0.012

^1.^ Analysis of covariance (ANCOVA) at admission adjusted for the age-adjusted Charlson comorbidity index; ANCOVA at discharge adjusted for the age-adjusted Charlson comorbidity index, Barthel Index at admission and length of stay. Abbreviations: BI: Barthel Index.

**Table 4 nutrients-12-02855-t004:** Physical performance according to nutritional status.

	Malnourished	Risk of Malnutrition	ANCOVA ^1^	Effect Ssize (η_p_^2^)
**Total sample**				
SPPB at admission	0.75 ± 1.56	1.53 ± 2.53	0.001	0.030
SPPB at discharge	4.06 ± 3.19	5.57 ± 3.38	0.004	0.027
POMA at admission	7.65 ± 8.04	10.17 ± 8.67	0.008	0.019
POMA at discharge	18.87 ± 7.1	21.11 ± 6.13	0.045	0.012
FAC at admission	0.71 ± 1.04	1.03 ± 1.24	0.011	0.017
FAC at discharge	2.81 ± 1.58	3.31 ± 1.45	0.063	0.011
**Orthogeriatric**				
SPPB at admission	0.45 ± 1.17	1.19 ± 2.05	0.005	0.035
SPPB at discharge	4.05 ± 2.84	5.78 ± 3.27	0.004	0.043
POMA at admission	6.7 ± 7.27	9.41 ± 8.23	0.027	0.021
POMA at discharge	19.35 ± 5.88	21.48 ± 5.71	0.142	0.010
FAC at admission	0.56 ± 0.89	0.91 ± 1.1	0.023	0.022
FAC at discharge	2.78 ± 1.38	3.4 ± 1.35	0.060	0.017
**Hospital-deconditioned**				
SPPB at admission	1.35 ± 2.02	2.14 ± 3.14	0.132	0.022
SPPB at discharge	4.56 ± 3.55	5.19 ± 3.29	0.934	<0.001
POMA at admission	10.57 ± 9.14	11.94 ± 9.6	0.458	0.005
POMA at discharge	19.77 ± 7.36	21.16 ± 6.27	0.318	0.012
FAC at admission	1.11 ± 1.25	1.34 ± 1.49	0.399	0.007
FAC at discharge	3.12 ± 1.72	3.33 ± 1.43	0.740	0.001

^1^ Analysis of covariance (ANCOVA) at admission adjusted for the age-adjusted Charlson comorbidity index; ANCOVA at discharge adjusted for age-adjusted Charlson comorbidity index, admission score and length of stay. Abbreviations: SPPB: Short Physical Performance Battery; POMA: Tinetti Performance-Oriented Mobility Assessment; FAC: Functional Ambulation Category.

**Table 5 nutrients-12-02855-t005:** Baseline characteristics of the sample according to falls.

	Total Sample		Orthogeriatric		Hospital-Deconditioned	
	Falls (*n* = 101)	No Falls (*n* = 274)	Falls (*n* = 62)	No Falls (*n* = 177)	Falls (*n* = 29)	No Falls (*n* = 75)
Age (years)	83.72 ± 6.37	82.6 ± 7.17	84.99 ± 6.01	83.75 ± 7.03	80.86 ± 6.87	80.73 ± 6.63
ACCI	6.28 ± 1.9	5.8 ± 1.85	6.11 ± 1.83	5.61 ± 1.87 *	6.34 ± 2.02	6.07 ± 1.76
MNA-SF	6.98 ± 1.74	7.6 ± 1.54 ***	6.9 ± 1.65	7.76 ± 1.6 ****	7.03 ± 2.03	7.21 ± 1.29
MMSE	19.49 ± 6.74	21.61 ± 6.38 **	18.69 ± 6.9	21.95 ± 6.04 ***	20.45 ± 6.49	21.53 ± 6.95
Barthel Index	32.96 ± 22.79	32.23 ± 23.15	29.45± 17.43	33.29 ± 22.66	42.03 ± 28.9	31.48 ± 23.09
SPPB	1.27 ± 2.36	1.16 ± 2.12	0.49 ± 1.23	1.02 ± 1.9 *	2.86 ± 3.26	1.29 ± 2.22 *
POMA	9.69 ± 8.46	8.84 ± 8.49	7.71 ± 7.31	8.52 ± 8.18	14.93 ± 9.17	9.80 ± 9.07 *
FAC	0.92 ± 1.18	0.88 ± 1.16	0.61 ± 0.89	0.82 ± 1.07	1.69 ± 1.42	1.04 ± 1.32 *

Abbreviations: ACCI: Age-adjusted Charlson comorbidity index; MNA-SF: Mini Nutritional Assessment Short Form; MMSE: Mini Mental State Examination; SPPB: Short Physical Performance Battery; POMA: Tinetti Performance-Oriented Mobility Assessment; FAC: Functional Ambulation Category. * *p* < 0.05; ** *p* < 0.01; *** *p* < 0.005; **** *p* < 0.001.

**Table 6 nutrients-12-02855-t006:** Logistic multivariate regression model of falling during rehabilitation.

	Total Sample ^1^		Orthogeriatric ^2^			Hospital-Deconditioned ^3^		
	OR (95%CI)	*p*		OR (95%CI)	*p*		OR (95%CI)	*p*
**ACCI**	1.148 (1.010–1.304)	0.034	**MMSE**	0.938 (0.892–0.987)	0.014	**SPPB**	1.236 (1.052–1.452)	0.010
**MNA-SF**	0.807 (0.696–0.936)	0.005	**MNA-SF**	0.806 (0.657–0.990)	0.040			

Abbreviations: OR: Odds Ratio; ACCI: Age-adjusted Charlson comorbidity index; MNA-SF: Mini Nutritional Assessment Short Form; MMSE: Mini Mental State Examination; SPPB: Short Physical Performance Battery; POMA: Tinetti Performance-Oriented Mobility Assessment; FAC: Functional Ambulation Category. ^1^ Estimates are based on *n* = 355 due to missing values; Variables in the model: age, ACCI, MNA-SF and MMSE. Hosmer–Lemershow goodness of fit, *p* = 0.792; Omnibus *p* < 0.001; R2 Nagelkerke = 0.056. ^2^ Estimates are based on *n* = 211 due to missing values; Variables in the model: age, ACCI, MNA-SF, MMSE and SPPB at admission; Hosmer–Lemershow goodness of fit, *p* = 0.881; Omnibus *p* < 0.001; R2 Nagelkerke = 0.108. ^3^ Estimates are based on *n* = 104; Variables in the model: Age, Barthel at admission, SPPB, POMA and FAC at admission; Hosmer–Lemershow goodness of fit, *p* = 0.405; Omnibus *p* < 0.001; R2 Nagelkerke = 0.093.

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
