# Peer review of "Nutritional Status Is Associated with Function, Physical Performance and Falls in Older Adults Admitted to Geriatric Rehabilitation: A Retrospective Cohort Study"

_nutrients, 2020, doi:10.3390/nu12092855_

Round 1

Reviewer 1 Report

Overall: Urquiza and colleagues conducted a retrospective cohort study of 375 older adults admitted to a geriatric rehabilitation facility to examine the association between nutritional status at admission (orthogeriatric or hospital deconditioning), the evolution of functional and physical outcomes, and the capability of nutritional status to identify fallers among these groups. This study addresses an important topic and it reads well. The study design/methodology along with the use of valid measures to assess selected outcomes could be viewed as strengths of the study. However, revisions are needed throughout the manuscript.

Introduction

  • The authors should provide a broader contextualization of nutritional status and its importance in the introduction. There are different approaches used to assess nutritional status and provide this vision to the readers is important.
  • The study hypothesis should be described in the introduction

Material and Methods

  • Please include the word “retrospective” before the “cohort” in the following sentence “This was a cohort study of medical records”
  • The authors mentioned that only patients admitted in the program for at least 25 days were included. Why 25? A rationale for that should be provided.
  • Page 3 – first paragraph. “The program started with joint mobilization and manual limbs… (i.e., canes or walkers).” I think it would be interesting to include this portion/information in a table for a better visualization purpose.
  • Data collection and outcomes

o   Nutritional status is the central point of the manuscript. I think it is better to describe it first in the paper instead of cognitive status.

o   Cognitive status. Authors should explain what the scores mean - is higher scores better?

o   Functional status. Authors should explain the scores and the questions.

o   How does fall information was collected? This should be described in the measures because it was an important variable in the analysis.

  • Statistical analysis

o   Did the authors have any missing data? How it was handled?

o   SPSS complete in-text reference must be provided. The following website can help: https://www.ibm.com/support/pages/how-cite-ibm-spss-statistics-or-earlier-versions-spss

  • Results

o   Authors should display effect sizes for the major analysis.

  • Discussion

o   The authors should develop a paragraph discussing their approach to assess nutritional status (i.e., MNA) compared to other methods, mainly touching in the idea with the pros and cons of their method compared to other methods. The following manuscript may be of a help: Shrivastava et al. Assessment of nutritional status in the community and clinical settings. Journal of Medical Sciences. 2014; 34(5): 211-213

o   In the last paragraph of the discussion the authors mentioned that the second part of the nutritional assessment was not completed (weight loss, BMI, etc.). However, in the outcomes section they report that they use the BMI item. This should be clarified.

Author Response

Overall: Urquiza and colleagues conducted a retrospective cohort study of 375 older adults admitted to a geriatric rehabilitation facility to examine the association between nutritional status at admission (orthogeriatric or hospital deconditioning), the evolution of functional and physical outcomes, and the capability of nutritional status to identify fallers among these groups. This study addresses an important topic and it reads well. The study design/methodology along with the use of valid measures to assess selected outcomes could be viewed as strengths of the study. However, revisions are needed throughout the manuscript.

Answer: Thank you for the detailed and constructive comments. We have addressed each one in the new version of the manuscript.

Introduction

  • The authors should provide a broader contextualization of nutritional status and its importance in the introduction. There are different approaches used to assess nutritional status and provide this vision to the readers is important.

Answer: We have included a new paragraph in the introduction section highlighting the early identification of malnutrition and describing different approaches to nutritional screening and assessment in older adults. It is highlighted on page 2, lines 55-64.

  • The study hypothesis should be described in the introduction

Answer: We have included the study hypothesis in the introduction section as suggested. It is highlighted on page 3, lines 97-99.

Material and Methods

  • Please include the word “retrospective” before the “cohort” in the following sentence “This was a cohort study of medical records”

Answer: We have made this change as suggested. It is highlighted on page 3, line 102.

  • The authors mentioned that only patients admitted in the program for at least 25 days were included. Why 25? A rationale for that should be provided.

Answer: To clarify this point, we have included additional information in the study design, setting, and participants´ section. According to the rehabilitation ward protocols, admitted patients had their second functional and physical outcome follow-up assessment at 25 days after admission, which is why we only included patients admitted to the ward for at least 25 days in the analysis. To prevent confusion, we have rewritten the sentence highlighted on page 3, lines 105-106, to specify that participation in the follow-up assessment was required for inclusion.

  • Page 3 – first paragraph. “The program started with joint mobilization and manual limbs… (i.e., canes or walkers).” I think it would be interesting to include this portion/information in a table for a better visualization purpose.

Answer: We have made this change as suggested and included a table with a better visualization of the rehabilitation program. It is highlighted on page 3, line 129.

  • Data collection and outcomes

o   Nutritional status is the central point of the manuscript. I think it is better to describe it first in the paper instead of cognitive status.

Answer: We agree with the reviewer’s comment and we have made this change as suggested. It is highlighted on page 4, lines 141-146.

o   Cognitive status. Authors should explain what the scores mean - is higher scores better?

Answer: We have included an explanation of the scores in the new version of the manuscript, as suggested by the reviewer. It is highlighted on page 4, line 150.

o   Functional status. Authors should explain the scores and the question.

Answer: We have explained the scores and parameters of each functional status test in the new version of the manuscript. It is highlighted on page 4, lines 152-153 and 160.

o   How does fall information was collected? This should be described in the measures because it was an important variable in the analysis.

Answer: We have added more information about the methodology of fall information collection in the new version of the manuscript. It is highlighted on page 5, lines 175-176.

  • Statistical analysis

o   Did the authors have any missing data? How it was handled?

Answer: We have specified that missing values were not included in the analysis in the statistical analysis section of the revised manuscript. It is highlighted on page 5, lines 194-195.

o   SPSS complete in-text reference must be provided. The following website can help: https://www.ibm.com/support/pages/how-cite-ibm-spss-statistics-or-earlier-versions-spss

Answer: We have made this change as suggested. It is highlighted on page 5, lines 198-199.

  • Results

o   Authors should display effect sizes for the major analysis.

Answer: We have included the partial eta squared calculation and reference values for effect size in the statistical analysis section (highlighted on page 5, lines 183-185). Additionally, in the results section, effect sizes are highlighted in Table 3,  page 6,  line 227 and Table 4, page 7, line 246.

  • Discussion

o The authors should develop a paragraph discussing their approach to assess nutritional status (i.e., MNA) compared to other methods, mainly touching in the idea with the pros and cons of their method compared to other methods. The following manuscript may be of a help: Shrivastava et al. Assessment of nutritional status in the community and clinical settings. Journal of Medical Sciences. 2014; 34(5): 211-213

 Answer: We agree with the reviewer that the comparison of our approach with other scales may be interesting for the readers of Nutrients. In this regard, we have included new information in two paragraphs of the discussion. First (page 9, lines 295-301), we mention that differences in malnutrition prevalence between different studies could depend on the screening tool used. Studies that used the MNA-SF had a similar prevalence of malnutrition to that in our study. However, studies using other tools found lower prevalence of malnutrition. Second (page 9, lines 312-321), considering the absence of predictive capability of the MNA-SF in hospital-deconditioned patients, we state that other scales that take into account the severity of the illness, inflammation or cachexia could be more accurate in predicting clinical outcomes in hospital-deconditioned patients.

o   In the last paragraph of the discussion the authors mentioned that the second part of the nutritional assessment was not completed (weight loss, BMI, etc.). However, in the outcomes section they report that they use the BMI item. This should be clarified.

Answer: We thank the reviewer for pointing out this mistake. We wanted to indicate that the full MNA was not performed. This includes some objective anthropometric measurements, that now are correctly specified the new version of the manuscript. This is highlighted on page 10, lines 385-387.

Reviewer 2 Report

Thank you for this very interesting paper on an important topic which can potentially help to identify patients at higher risk of being a faller in clinical setting. I have a few suggestions which I think will improve the quality of this manuscript:

  1. I suggest you add a small section in the introduction explaining why it is important to compare malnutrition status and risk of being faller according to the reason of the patient’s admission (i.e. orthogeriatric vs. hospital-deconditioned). Because I feel when reading this concept first occurred in the aim of your study in line 81 without explanation.
  2. Following the previous comment, why did you choose to compare between these 2 reasons for admission (orthogeriatric vs. hospital-deconditioned). Why not identify conditions that are directly related to lower-limb function (e.g. hip-replacement, lower-limb surgery, fractures, deconditioning etc..which likely will lead to more fallers) vs. those without these conditions? I am not suggesting to re-write the whole manuscript, but just emphasize a bit more why you chose to do it this way?
  3. Retrospectively, your results indicate that among those admitted due to hospital-deconditioned, you could not see association between nutritional status and risk of fall (probably due to a flooring effect, i.e. admitted with poor nutrition status). Could you find any literature on this group of patients and could maybe describe why it is important clinically to distinguish this patient group with the acute orthopedic geriatric patients?
  4. In the discussion section, you hypothesized that “hospital-deconditioned patients with a higher score on the SPPB might be more independent for displacements and consequently, have a higher chance for suffering falls than patients with poor physical performance “. However, if we look at Table 3, the SPPB score at admission and at discharge are actually higher in those with hospital-deconditioned compared to those with orthogeriatric conditions for the malnourished group (1.35 ± 2.02 vs. 0.45 ± 1.17 at admission; 4.56 ± 3.55 vs. 4.05 ± 2.84). So why is that the association between SPPB score and fall risk is counter-intuitive among the deconditioned patients but not the orthogeriatric patients?  I think you need to be careful when writing it into the conclusion that “in hospital-deconditioned patients, a higher score on the SPPB was an independent factor for identifying fallers “. I think this only is true when the SPPB is within a very low range, so this needs to be specified. Otherwise, readers could be misled into thinking a higher SPPB (better function) leads to higher fall risks in general.

Minor comment:

  1. (Table5): You can just report OR (95% CI), same as you reported in lines 243-249 instead of exp beta (95% CI). And I don’t think the separate beta value is needed.

Author Response

Thank you for this very interesting paper on an important topic which can potentially help to identify patients at higher risk of being a faller in clinical setting. I have a few suggestions which I think will improve the quality of this manuscript:

Answer: Thank you for the positive and constructive comments. We have addressed each one in the new version of the manuscript.

  1. I suggest you add a small section in the introduction explaining why it is important to compare malnutrition status and risk of being faller according to the reason of the patient’s admission (i.e. orthogeriatric vs. hospital-deconditioned). Because I feel when reading this concept first occurred in the aim of your study in line 81 without explanation.

Answer: We have included further explanation in the introduction section as suggested by the reviewer. In the new version of the manuscript, taking into account the ESPEN criteria, we have described the most frequent differences in malnutrition etiology. Patients admitted due to chronic conditions (hospital-deconditioned) were undergoing disease-related malnutrition with previous inflammatory response as well as muscle and weight loss. In contrast, in acute patients (orthogeriatric), malnutrition was related to their acute injury with a strong and acute inflammatory response. This is highlighted on page 2, lines 85-90.

  1. Following the previous comment, why did you choose to compare between these 2 reasons for admission (orthogeriatric vs. hospital-deconditioned). Why not identify conditions that are directly related to lower-limb function (e.g. hip-replacement, lower-limb surgery, fractures, deconditioning etc., which likely will lead to more fallers) vs. those without these conditions? I am not suggesting to re-write the whole manuscript, but just emphasize a bit more why you chose to do it this way?

Answer: While what the reviewer proposes could be a different and interesting approach to analyze this population, the sample was not big enough for further segmentation and analysis depending on lower-limb pathology. We further explained our resoning for comparing between orthogeriatric and hospital-deconditioned patients in the introduction section of the revised manuscript as described in our response to comment #1 above.

  1. Retrospectively, your results indicate that among those admitted due to hospital-deconditioned, you could not see association between nutritional status and risk of fall (probably due to a flooring effect, i.e. admitted with poor nutrition status). Could you find any literature on this group of patients and could maybe describe why it is important clinically to distinguish this patient group with the acute orthopedic geriatric patients?

 Answer: Cederholm et al. (2017) explained that the etiology of malnutrition is different depending on the acute or chronic nature of the disease. Chronic patients usually suffer malnutrition associated with cachexia. However, acute patients have highly increased pro-inflammatory activity, which may cause a fast decline of body energy and nutrient stores Therefore, we consider it adequate to distinguish between hospital-deconditioned (chronic) and orthogeriatric (acute) patients. This issue is mentioned in the introduction and discussion of the new version of the manuscript. It is highlighted on page 2, lines 85-90 (introduction) and page 9, lines 312-321 (discussion).

  1. In the discussion section, you hypothesized that “hospital-deconditioned patients with a higher score on the SPPB might be more independent for displacements and consequently, have a higher chance for suffering falls than patients with poor physical performance “. However, if we look at Table 3, the SPPB score at admission and at discharge are actually higher in those with hospital-deconditioned compared to those with orthogeriatric conditions for the malnourished group (1.35 ± 2.02 vs. 0.45 ± 1.17 at admission; 4.56 ± 3.55 vs. 4.05 ± 2.84). So why is that the association between SPPB score and fall risk is counter-intuitive among the deconditioned patients but not the orthogeriatric patients?  I think you need to be careful when writing it into the conclusion that “in hospital-deconditioned patients, a higher score on the SPPB was an independent factor for identifying fallers “. I think this only is true when the SPPB is within a very low range, so this needs to be specified. Otherwise, readers could be misled into thinking a higher SPPB (better function) leads to higher fall risks in general.

 Answer: As suggested by the reviewer, we have added further explanation to the conclusion section to prevent readers from misunderstanding that better function in hospital-deconditioned patients leads to higher fall risk. This is highlighted on page 11, lines 400-401

Minor comment:

  1. (Table5): You can just report OR (95% CI), same as you reported in lines 243-249 instead of exp beta (95% CI). And I don’t think the separate beta value is needed.

Answer: We have made this change as suggested by the reviewer. It is highlighted on page 8, line 275, Table 6.